# Enhanced Magnetoimpedance Effect in Co-Based Micron Composite CoFeNiSiB Ribbon Strips Coated by Carbon and FeCoGa Nanofilms for Sensing Applications

**DOI:** 10.3390/s24102961

**Published:** 2024-05-07

**Authors:** Zhen Yang, Mengyu Liu, Jingyuan Chen, Xuecheng Sun, Chong Lei, Yuanwei Shen, Zhenbao Wang, Mengjiao Zhu, Ziqin Meng

**Affiliations:** 1Guangxi Key Laboratory of Brain-Inspired Computing and Intelligent Chips, School of Electronic and Information Engineering, Guangxi Normal University, Guilin 541004, China; 1349375154@stu.gxnu.edu.cn (M.L.); cjy@stu.gxnu.edu.cn (J.C.); syw@stu.gxnu.edu.cn (Y.S.); 1354724210@stu.gxnu.edu.cn (Z.W.); badzmj@stu.gxnu.edu.cn (M.Z.) meng@stu.gxnu.edu.cn (Z.M.); 2Key Laboratory of Integrated Circuits and Microsystems (Guangxi Normal University), Education Department of Guangxi Zhuang Autonomous Region, Guilin 541004, China; 3Institute for Health Innovation and Technology, National University of Singapore, Singapore 117599, Singapore; 4Microelectronic Research & Development Center, School of Mechatronics Engineering and Automation, Shanghai University, Shanghai 200444, China; sunxc@shu.edu.cn; 5National Key Laboratory of Advanced Micro and Nano Manufacture Technology, Department of Micro-Nano Electronics, School of Electronic Information and Electrical Engineering, Shanghai Jiao Tong University, Dongchuan Road 800, Shanghai 200240, China; leiqhd@sjtu.edu.cn

**Keywords:** magnetoimpedance effect, composite ribbon strips, nanofilms, sensing applications

## Abstract

Quenched Co-based ribbon strips are widely used in the fields of magnetic amplifier, magnetic head material, magnetic shield, electric reactor, inductance core, sensor core, anti-theft system label, and so on. In this study, Co-based composite CoFeNiSiB ribbon strips with a micron width were fabricated by micro-electro-mechanical systems (MEMS) technology. The carbon and FeCoGa nanofilms were deposited for surface modification. The effect of carbon and FeCoGa nanofilm coatings on the crystal structure, surface morphology, magnetic properties, and magnetoimpedance (MI) effect of composite ribbon strips were systematically investigated. The results show that the surface roughness and coercivity of the composite ribbon strips are minimum at a thickness of the carbon coating of 60 nm. The maximum value of MI effect is 41% at 2 MHz, which is approximately 2.4 times greater than plain ribbon and 1.6 times greater than FeCoGa-coated composite ribbon strip. The addition of a carbon layer provides a conductive path for high frequency currents, which effectively reduces the characteristic frequency of the composite ribbon strip. The FeCoGa coating is able to close the flux path and reduce the coercivity, which, in turn, increases the transverse permeability and improves the MI effect. The findings indicate that a successful combination of carbon layer and magnetostrictive FeCoGa nanofilm layer can improve the MI effect and magnetic field sensitivity of the ribbon strips, demonstrating the potential of the composite strips for local and micro area field sensing applications.

## 1. Introduction

Among various magnetoimpedance (MI) materials, Co-based amorphous ribbons have attracted considerable attention due to their higher permeability, lower saturation field, lower power dissipation, lower hysteresis, and tunable magnetic anisotropy [1]. In addition, they are not sensitive to mechanical stress and can work in harsh environments for a long time, so the stability and reliability of product devices are greatly improved, especially for military products sensing applications [2]. At present, the research on enhanced MI properties of Co-based amorphous ribbon mainly focuses on annealing process, optimizing structural parameters, and controlling the surface [3,4,5]. However, higher temperature annealing treatment (>350 °C) on the ribbons will lead to a certain degree of brittleness, which greatly limits its application. Meander MI elements have become a focus of structural optimization research. Many theoretical and experimental studies on enhanced MI effects in ribbon meanders have been reported [6,7,8,9]. Laser cutting and wet etching are the main fabrication methods for micro-patterned ribbons [10,11]. However, laser cutting also comes with a large amount of heat generation, which affects the smoothness of ribbons and might modify the domain structure of the micro-ribbon, thus limiting the MI performance of the micro-ribbon. Although the MI properties of ribbon meanders prepared by wet etching technology are better than those of ribbon strips, there is a large line etching error during the process of etching due to the smaller spacing between the lines [12]. Therefore, ribbon strips have certain advantages in practical applications compared with ribbon meanders, as the ribbon strips are easier to fabricate.

Recently, many efforts have been devoted to enhancing the MI amplitude of ribbon strips by controllable modifications on the ribbon surface. Various coating methods have been adopted. Oxide nanofilms (ZnO, CuO, ~450 nm) and diamagnetic organic nanofilms have been deposited on the surface of 2705 Co-ribbon (width = 5 mm) [13,14,15], resulting in a significant enhancement of the MI effect. The enhanced MI response of a Co-based ribbon (width = 2 mm) coated with Co (50 nm), CoFe (400 nm), and CoFe_2_O_4_ (85 nm) have been reported [16,17,18]. The carbon nanomaterials (carbon nanotubes, CNT, and graphene oxide) have also been used to modify the surface of the Co-ribbon for the enhancement of the MI [19,20]. Due to the fragility of Fe-based ribbons themselves, Fe-based composite ribbon strips are less effective in obtaining a higher MI performance. NiZn (60 nm) and TiO_2_ (100 nm) nanofilms have been used to modify Fe-based ribbon strips (width, 0.6–5 mm) and have been proved effective in enhancing MI properties [21,22]. The magnetic properties and inter-layer coupling of Fe-based composite ribbon strips have also been investigated. In order to analyze the change in MI curves as a result of different thicknesses of coatings, the effect of various thicknesses of Fe_50_Pt_50_, FePd, IGZO, and FeGa on the surface of Fe-based ribbons (width, 0.6–3 mm) by magnetron sputtering have been evaluated [23,24,25,26]. Research results show that there is an optimal thickness of coatings for the best MI effect.

In order to effectively obtain the highest possible MI effect, sandwiched composite ribbon strips are required [27]. In a sandwich configuration, the conductive layer serves as the primary pathway for AC conduction. A smaller resistance of the inner conductive layer makes the inductance generated by the outer soft magnetic layer greater than that of the inner conductive layer. This is beneficial to improve the MI effect of composite ribbons [28]. Therefore, discovering a substance with elevated electrical conductivity in the form of a sandwich structure is important for the development of MI sensors with superior performance. A novel FINEMET/rGo/FeCo sandwich composite ribbon was proposed by Y. J. Chen et al. [29]. The rGo layer can effectively reduce the surface roughness of Fe-based ribbons and the closed flux path. The magnetic exchange interaction and surface stress between layers due to a lattice match between the FeCo layer and FINEMET ribbon was perceived as the main reason behind the enhancement of the MI effect. In addition, the MI ratios in FePd/FINEMET/FePd, FINEMET/SiO_2_/FePd, and FINEMET/IGZO/CoFeSiB sandwiched structures (Width_FINEMET_ = 0.6 mm) have been systematically investigated [30,31]. The dipole interaction and magnetic interaction mechanisms have been adopted to explain the results [32]. There is significant exchange coupling between the magnetic moments of its electron spin due to the unique atomic structure of FeCoGa and the magnetic moment properties of electron spin [33]. The magnetostrictive FeCoGa films are widely used in electronics, communication, energy, and other fields, with the advantages of high magnetic permeability, high magnetic induction strength, high corrosion resistance, etc. Carbon nanofilms have been widely used in the fields of mechanics, thermology, biology, and material preparation due to their high hardness, high wear resistance, high thermal conductivity, high corrosion resistance, low coefficient of friction, and good biocompatibility [34,35]. Therefore, the skillful use of carbon and FeCoGa nanofilm coatings to improve MI is very effective.

In almost all of the abovementioned composite ribbon strips studies, the ribbon strips have been obtained by trialing millimetric widths for experimental purposes. A larger line width error is produced by manual cutting, and the micrometer width sample cannot be obtained by manual cutting. The minimum width of ribbon strips prepared by the melt spinning method is also 0.6 mm [23]. In the study of ribbon meanders, the smallest line widths can be controlled down to hundreds or even tens of microns by MEMS technology. In the ribbon meanders, there exists an optimum ratio among the dimensional parameters of length, width, and line gap of the sample for the highest MI effect. However, the line width is negatively correlated with the MI effect for ribbon strips. With the development of science and technology, electronic devices are becoming smaller and smaller, and packages are becoming more and more miniaturized. Therefore, it is important to fabricate the Co-based composite ribbon strips with micro width for a better MI effect. The composite ribbon strips with a width of tens of microns are expected to be used in more industrial applications as amorphous wires due to their superior MI performance. In this work, carbon- and FeCoGa-nanofilms-coated Co-based composite ribbon strips with a 300 μm width were prepared by MEMS technology. A carbon nanolayer with different thicknesses and FeCoGa nanolayers with constant thickness were designed. The crystalline structure, surface morphology, and magnetic properties of composite ribbon strips are investigated. The principle of enhanced GMI effect in composite ribbons are explained.

## 2. Materials and Methods

In this context, commercial Co-based ribbons with Co_82.6_Fe_4.2_Ni_1.2_Si_8.8_B_3.2_ nominal composition were purchased from Jiangxi Dayou Scientific & Technical Co., Ltd., Jiangxi, China. The thickness and width of these ribbons are 20 μm and 12 mm, respectively. The main magnetic properties of the ribbon are as follows: maximum permeability μ_max_ is 4 × 10^5^, resistivity ρ is 10 μΩ·cm, and Curie temperature is 250 °C. The saturation induction B_S_ of the ribbon is 0.58 T and the coercivity H_c_ is 0.3 A m^−1^. No additional heat treatment of the Co-based ribbon was performed. Compared to the ribbon adopted in previous reports [6,36], the amorphous ribbon utilized in this design is not only superior to Metglas and Hebei King Do in some key parameters (such as Curie temperature, coercive force and crystallization temperature), but it is also cheaper and easy to purchase. The physical and magnetic properties of the ribbons are presented in Table 1. In subsequent works, the MI performance can be substantially improved after a certain degree of annealing (higher Curie temperature).

The structure parameters of composite ribbon strips were designed to be 300 µm wide. For ease of measurement, the length of composite ribbon strips was chosen to be 5 mm. The micron composite ribbon strips were obtained via designed mask and fabricated by MEMS technology including photolithography, wet etching, and magnetron sputtering. Detailed steps for the patterned single-layer ribbon strip have been reported elsewhere [8]. After finishing the patterned single-layer ribbon strip, the photoresist with a thickness of 1 μm was spin-coated on the surface of the ribbon strip. An anti-structured mask was used for the second photolithography process to obtain a sputtering zone of patterned ribbon strip. Then, the carbon and FeCoGa layers were sputtered sequentially. The different thicknesses of the carbon nanolayer and the constant FeCoGa nanofilm were deposited. The base pressure was 4.4 × 10^−5^ Pa, and the Argon gas pressure was 1.8 Pa during sputtering. The magnetron sputtering power was 150 W and the sputtering rate of carbon and FeCoGa was about 0.76 nm/min and 8.83 nm/min, respectively. The carbon nanolayer with a thickness of 0 nm, 30 nm, 60 nm, 90 nm, 120 nm, and 150 nm (t_c_ = 0–150 nm), and the FeCoGa nanolayer with a thickness of 220 nm were sputtered. After sputtering, composite ribbon strips (t_c_ = 0 nm, ribbon + FeCoGa; t_c_ = 30–150 nm, ribbon + carbon + FeCoGa) were obtained.

The crystal structure of the composite ribbon was examined by X-ray diffraction (XRD) analysis under Cu–Kα radiation with a scan range of 30–90° and a scan rate of 2°/min. Plan view and cross-section images of the composite ribbon were obtained using a field-emission scanning electron microscope (FESEM, CARL-ZEISS SMT). The surface roughness of the composite ribbon was measured using an atomic force microscope (AFM, NX20). The scanning area was 10 µm × 10 µm. Hysteresis loops were measured at room temperature with a vibrating sample magnetometer (VSM, Quantum Design SQUID-VSM (MPMS-3)). The GMI effect of the specimen was tested using a 4294A impedance analyzer (Keysight) with an alligator clip and the 16047D accessory. The AC frequency ranged from 100 Hz to 30 MHz. A Helmholtz coil system was used to create a uniform magnetic field from −140 Oe to 140 Oe along the ribbons’ longitudinal axis. A schematic diagram of the measurement is shown in Figure 1. The GMI ratio of the composite ribbon can be detected by changing the value of the applied DC magnetic field. The GMI ratio and the magnetic field sensitivity (η) are defined as:MI(%) = 100% × (Z(H) − Z(H_max_))/Z(H_max_)(1)
η = d (∆Z/Z)/d(H)(2)
where Z(H) refers to the impedance value in a specific external magnetic field and Z(Hm_ax_) refers to the impedance when the maximum external magnetic field is applied.

## 3. Results and Discussion

Figure 2 shows the X-ray diffraction patterns of the as-cast ribbon strip and of the composite ribbon strip with different carbon layer thicknesses. The XRD patterns of the as-cast ribbon strip demonstrate amorphous behavior, with a board hump peak at an angle of around 45°. As the thickness of carbon increases, the peak value of the composite ribbon correspondingly increases. However, for all deposited ribbons, it can be found that XRD patterns have no obvious crystal peaks, and each diffraction peak is wide and low. This indicates that the carbon films of different thicknesses deposited on the Co-based ribbon were amorphous. The SEM surface morphologies of the as-cast ribbon and of the composite ribbons with different thicknesses of the carbon layer were observed, as shown in Figure 3. It can be seen from the SEM image in Figure 3b that the deposited FeCoGa film had a flat morphology apart from a few imperfections on the surface, indicating that the FeCoGa film prepared by magnetron sputtering had a good densification and uniformity. In addition, it is evident that the carbon and FeCoGa nanolayers were well compounded on the surfaces of the Co-based ribbon. The carbon nanofilm on the surface of the composite ribbon with a thickness of 60 nm was dense and smooth, as shown in Figure 3c. When the thickness of the carbon on the composite ribbon strip was 120 nm, obvious particle-like agglomerates on the surface of the composite ribbon strip were observed, possibly formed by some of the carbon agglomerates. From Figure 3e, it can be seen that the cross section of the composite ribbon strip has an obvious layered structure, effectively proving that high-quality carbon and FeCoGa nanofilms were successfully compounded with the ribbons. The carbon coating layer maintained excellent electrical conductivity, facilitating the passing of current at high frequencies which increases the rate of change for MI.

The microscopic morphology of the composite ribbon strips is shown in Figure 4. It was found that the roughness (Ra) of the composite ribbon strips decreased first and increased afterward with increasing carbon film thickness. As the thickness of the carbon nanofilm increased, the peaks and valleys tended to decrease. When the carbon layer was 60 nm, the surface of the composite ribbons was flat and the Ra was at its minimum. As the carbon layer thickness increased, it significantly affected the surface structure of the composite ribbons, inducing an increase in Ra on the surface of the samples. The coating layers roughness has significant effects on magnetic properties. If the magnetic flux of the surface cannot be guided completely within the magnetic circuit, this will lead to the generation of stray magnetic fields (H_s_) [13,16]. Therefore, the smaller the surface roughness of the composite ribbon strip, the smaller the magnetic loss and the larger the transverse permeability, which is conducive to the enhancement of the MI effect.

Figure 5 shows the normalized magnetic hysteresis loops of the as-cast ribbon strip and composite ribbon strips. It was found that all curves had an S-shaped profile over the applied magnetic field at room temperature, indicating that all samples have excellent soft magnetic properties. As displayed in Figure 4b, the composite ribbon strips exhibited a better transverse magnetic structure in comparison to the other samples.

Moreover, all of the composite ribbon strips revealed narrower curves and were found to be saturated at lower magnetic fields. Since the thickness of the Co-based ribbon was much larger than that of the nanofilm coatings, the ribbon played an important role in the magnetization of composite samples. The FeCoGa and the carbon nanolayer can effectively reduce the H_c_ of the as-cast ribbon. The H_c_ of the composite ribbon strips decreased and then increased with the increase in thickness of the carbon layer, as shown in the inset. When the carbon layer thickness was 30 nm, 60 nm, 90 nm, 120 nm, and 150 nm, the H_c_ was 4.8 Oe, 3.6 Oe, 2.5 Oe, 3.2 Oe, 4.3 Oe, and 6.5 Oe, respectively. The as-cast ribbon exhibited the highest coercivity of 7.9 Oe. Therefore, the addition of FeCoGa and carbon coating layers resulted in reduced leakage flux, improved magnetic circuit closure, diminished surface roughness, and increased permeability and magnetizing strength [16]. As the carbon layer’s thickness increased, the thin film coating exerted stress onto the ribbon’s surface, changing the domain structure and increasing the roughness, resulting in a decrease in soft magnetic properties [15]. The above result was consistent with the conclusions drawn from SEM and AFM analyses.

Figure 6 shows the magnetic field dependence of the MI curves of the as-cast ribbon and composite ribbons with varying thicknesses of the carbon layer at different frequencies. It is obvious that all MI curves exhibit a double-peak shape in the magnetic field range above 1 MHz. This suggests that all samples possessed a transverse magnetic anisotropy. When the external magnetic field (H_ex_) was close to the magnetic anisotropy field (H_k_), the maximum MI value was reached. The shape of a sample’s magnetic domain is quite related to its magnetic moment orientation. When H_ex_ approached H_k_, the orientation of the magnetic moment in the sample became unstable and susceptible to the alternating driving field. In this case, the transverse permeability of the sample reached the maximum value, causing the largest MI. As evident from Equations (1) and (2), the MI ratio and sensitivity of the as-cast ribbon were only 17.1% and 6.5%/Oe at a frequency of 3 MHz. The maximum MI ratio increased first and then decreased as the carbon layer thickness increased. When the FeCoGa coating layer was added, the impedance ratio improved to 25.5%, while sensitivity slightly decreased (6.38%/Oe). This is due to the addition of the FeCoGa layer changing the domain structure of the as-cast ribbon, causing a shift in H_k_ to higher fields. When the thickness of the intermediate layer (carbon coating layer) was 60 nm, both the MI ratio and the sensitivity reached their maximum values, which were 41% and 20.5%/Oe, respectively. This indicates that the MI effect and the sensitivity of the composite ribbon can be effectively increased by inserting a suitable carbon layer thickness.

There are two interaction mechanisms that occur in composite ribbons with different thicknesses of the carbon coating layer: exchange coupling (H_exch_) and dipole interaction (H_dip_). With a gradual increase in the carbon layer’s thickness, the exchange coupling effect becomes weaker and dipole interactions are gradually revealed. The H_dip_ between the FeCoGa coating layer and the Co-based ribbon can be obtained using Equation (3):H_dip_ = −KM_FeCoGa_(3)
where M_FeCoGa_ denotes the magnetic moment of the FeCoGa coating layer and K is the geometric factor, which can be expressed as:K = L/(r^2^ + L^2^)^3/2^(4)
where L is length and r is the distance between the Co-based ribbon and the FeCoGa coating layer. Therefore, it can be understood that the dipole interaction depends on r.

As shown in Figure 7a, the peak field (H_p_) of the composite ribbons decreased with increasing thickness of the carbon layer. The H_p_ can be expressed as [28]
H_p_ =H_p(Co)_ + H_dip_ + H_stress_(5)
where H_p_ represents the peak field of the composite ribbon strips, H_p(Co)_ refers to the peak field of the Co-based ribbon, and H_stress_ is the stress energy of the coating layer. The H_dip_ changes as a function of carbon layer thickness, as shown in Figure 7b. It can be seen that the dipole field of composite ribbons gradually decreased as the carbon layer thickness increased. From Equation (4), it can be seen that, as the thickness of the carbon layer gradually increases, the distance (r) between the as-cast ribbon and the FeCoGa thin film also increases. Then, the K decreases, resulting in a reduction of the dipole field. Therefore, the addition of the intermediate carbon layer can effectively regulate the MI effect and the dipole field of the composite ribbon.

Figure 8a displays the frequency dependence of the MI ratios for both the as-cast ribbon and the composite ribbon with varying thicknesses of the carbon layers. The MI ratio first increased and then decreased with increasing frequency over the entire frequency range, reaching the maximum MI ratio value at the characteristic frequency (f_max_). This can be explained by the contribution of the domain wall movement and magnetization rotation to the effective permeability at different frequencies. In the low frequency range, magnetic induction is dominant, leading to a weaker skin effect and thus a lower MI effect. As the frequency increases to the intermediate frequency, the motion of the domain walls and magnetization enhances the effective permeability, which improves the skinning depth of the sample and results in the enhanced MI effect. However, as the frequency increases further, the damping of the eddy currents gradually increases, causing a reduction in the effective permeability and resulting in a decrease in the MI effect. Figure 8b illustrates variations in the maximum MI ratio and the characteristic frequency (f_max_) according to the thickness of the carbon layer. As it can be seen, the maximum MI ratio of the composite ribbon showed an “increase first and then decrease” trend with increasing carbon thickness. The maximum MI ratio reached was 41% when the carbon layer thickness was 60 nm, such a value being 2.4 times that of the as-cast ribbon (17.1%) and 1.6 times that of the ribbon/FeCoGa composite ribbon (25.5%). Composite ribbons containing carbon layers allow high-frequency drive current to pass through, reducing the impact of AC resistance and skin depth on the sample. The MI ratio dropped to 25.2% when the carbon layer thickness was 150 nm. This is because when the carbon layer reaches a particular depth, the composite ribbon surface roughness increases and the resulting H_s_ gradually increases, affecting the magnetic circuit closure, increasing the magnetic loss, and reducing the MI effect. Furthermore, as the thickness of the carbon layer increases, the dipole interaction gradually decreases and the magnetic properties deteriorate, phenomena which also contribute to the worse performance of MI. The f_max_ (the frequency at which the MI ratio reaches the maximum value) is an important index reflecting the MI effect, and it is also an important parameter affecting low-frequency application of the MI sensor. It can be expressed as follows:f_max_ = 1/μ_φ_πt^2^σ(6)
where μ_φ_ and σ represent the saturated magnetic permeability in a 140 Oe field and electric conductivity of the ribbon, respectively, and t is the thickness of the coating layer. It can be seen from Equation (3) that, when the μ_φ_ and σ parameters are fixed, the characteristic frequency of the composite ribbons gradually decreases as the thickness of the carbon layer increases. These calculations are well consistent with our experimental results. The frequencies were 4 MHz and 2 MHz for the ribbon/FeCoGa ribbon and ribbon/carbon/FeCoGa (t_c_ = 150 nm), respectively. This shows that the thickness of the carbon coating layer can effectively reduce the working frequency of composite ribbons. Therefore, the appropriate carbon coating layer thickness can effectively enhance the MI response and reduce the characteristic frequency. This optimization is especially significant for biological applications of the MI sensor.

Narrower strips (thin film or ribbon) for MI elements are beneficial to reduce the demagnetizing effect and improve the MI effect [37]. The demagnetizing field in the direction of the width hinders the control of the anisotropy of the elements [38]. To contribute to the development of more miniaturized MI-type sensors, we will design a smaller width for ribbon strips in future work. However, considering the line error of the ribbon strip when using the wet etching technique (25–35 μm), ribbon strips with a width of <100 μm are expected to be obtained by wet etching. The more miniaturized MI-type sensor with a better MI performance will be used in more sensing fields.

## 4. Conclusions

In this work, carbon- and FeCoGa-nanolfilms-coated Co-based composite ribbon strips that were 300 μm wide were successfully fabricated by using the MEMS process. By comparing the crystal structure, surface roughness, magnetic properties, and the MI effect of the composite ribbon strips, the results show that a thickness of 60 nm for the carbon layer led to the composite ribbon strip having the smallest roughness (9.78 nm) and the best magnetic properties (the coercive force is 2.5 Oe), as well as an MI ratio of 41% at 2 MHz. The carbon coating layer can provide conductive pathways to high-frequency currents, which, in turn, reduce the characteristic frequency of composite ribbons. The addition of a FeCoGa layer has the effect of closing the magnetic flux path runway and reducing the magnetic leakage. These results can provide a new direction and theoretical foundation for improving the MI effect in the soft magnetic composite ribbon strips used in the local and micro area field of sensing applications.

## Figures and Tables

**Figure 1 sensors-24-02961-f001:**
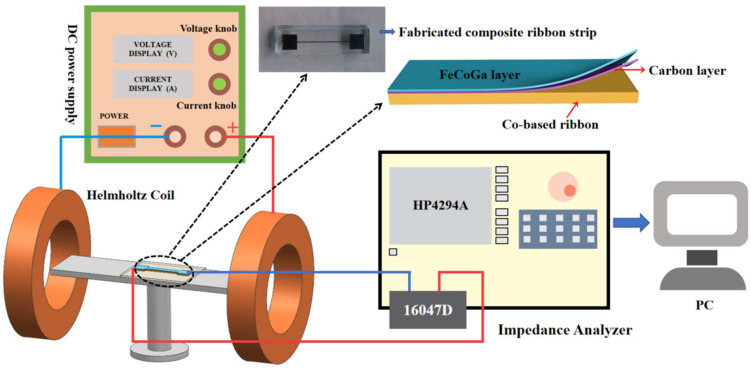
The schematic diagram of the measurement for MI effect (inset: fabricated composite ribbon strip).

**Figure 2 sensors-24-02961-f002:**
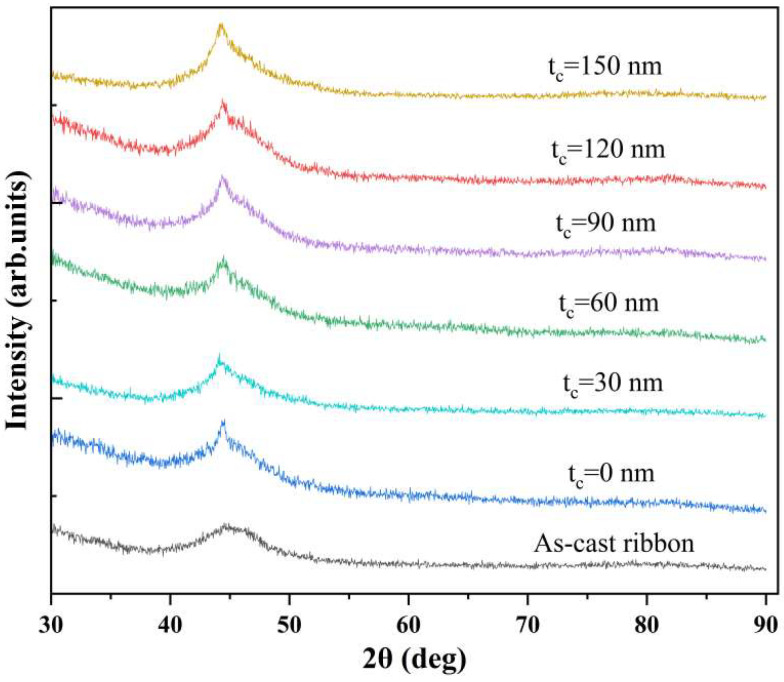
XRD patterns of the Co-based ribbon and of the composite ribbon strips with different thicknesses of the carbon coatings.

**Figure 3 sensors-24-02961-f003:**
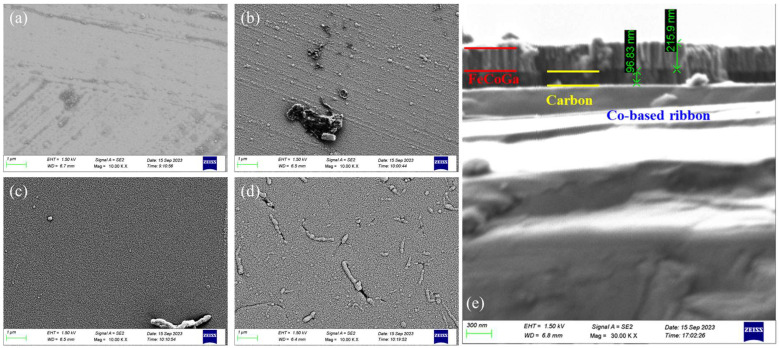
Typical SEM images: (**a**) the Co-based ribbon, (**b**–**d**) composite ribbon strips at t_c_ = 0 nm, 60 nm, and 120 nm; (**e**) cross section of the composite ribbon at t_c_ = 90 nm.

**Figure 4 sensors-24-02961-f004:**
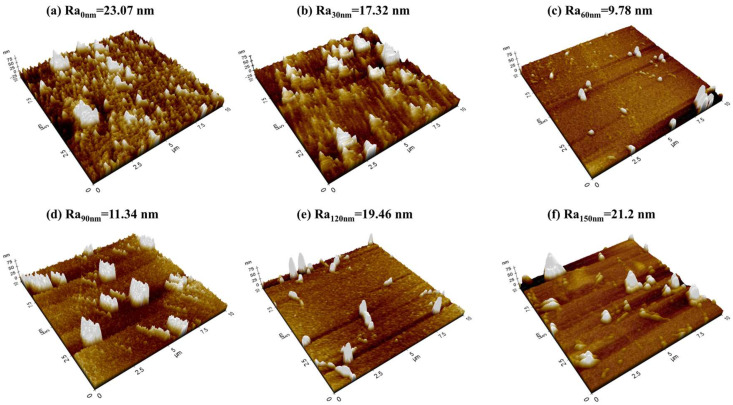
AFM diagram of composite ribbon strips with different carbon layer thickness: (**a**) t_c_ = 0 nm, (**b**) t_c_ = 30 nm, (**c**) t_c_ = 60 nm, (**d**) t_c_ = 90 nm, (**e**) t_c_ = 120 nm, and (**f**) t_c_ = 150 nm.

**Figure 5 sensors-24-02961-f005:**
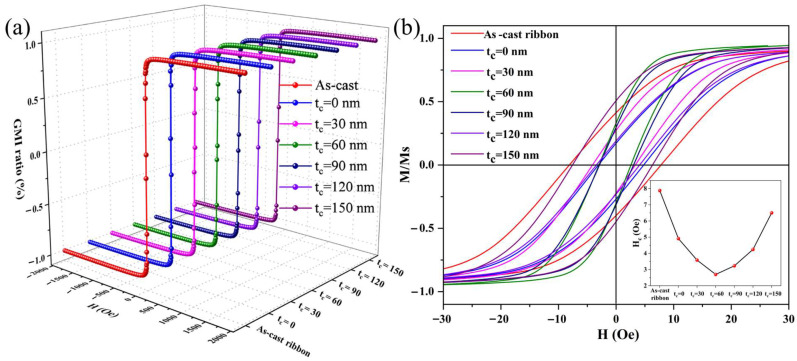
Normalized magnetic hysteresis loops of the as-cast ribbon and composite ribbon strips (**a**), and partial enlargement of the hysteresis loops (**b**). The insert shows the variation curve of H_c_ with different carbon layer thicknesses.

**Figure 6 sensors-24-02961-f006:**
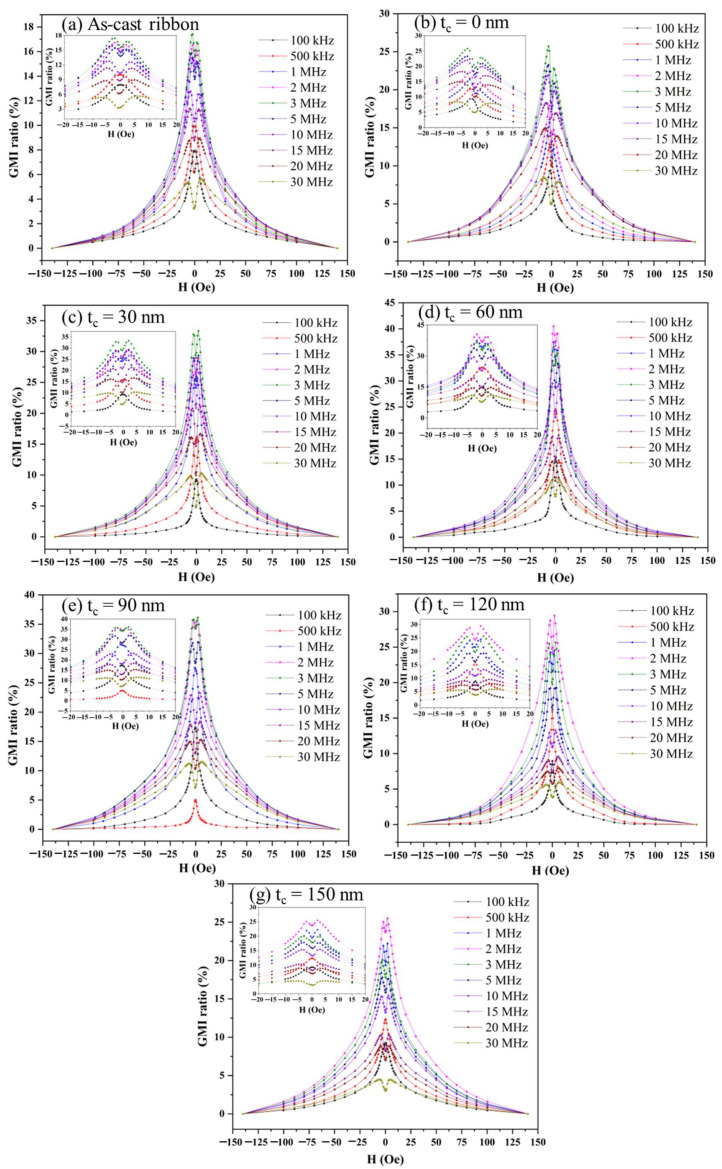
Magnetic field dependence of the MI ratio of composite ribbon strips with different carbon layer thicknesses at different frequencies.

**Figure 7 sensors-24-02961-f007:**
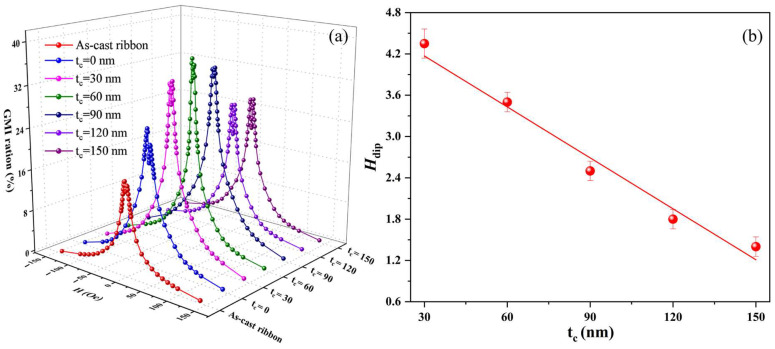
(**a**) Magnetic field dependence of MI ratio of the ribbon strips with different carbon layer thicknesses at 3 MHz. (**b**) H_dip_ variation curves with varying carbon layer thickness.

**Figure 8 sensors-24-02961-f008:**
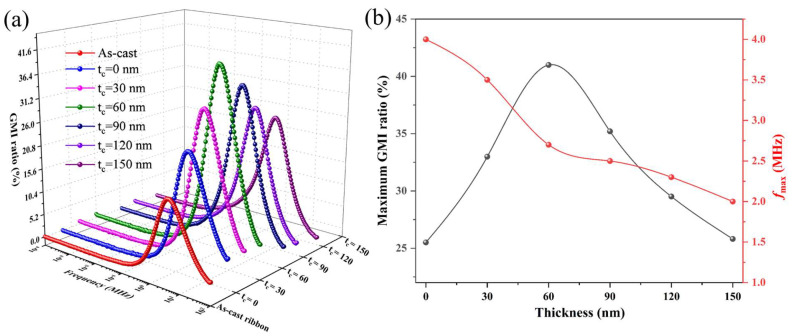
(**a**) Frequency dependence of the MI ratio with different carbon layer thicknesses; (**b**) variation of the maximum MI ratio and f_max_ with different thicknesses of the carbon layer.

**Table 1 sensors-24-02961-t001:** Magnetic and physical properties of three commercial ribbons.

Magnetic and Physical Properties			
Brand Name	Hebei King Do	Metglas	Jiangxi Dayou
Saturation induction (T)	>0.55	0.5	0.58
Curie temperature (°C)	205	200	250
Maximum permeability (µ)	>1,200,000	1,000,000	>1,200,000
Coercive force (A/m)	<2.0	2	0.3
Density (g/cm^3^)	8.5	7.59	7.25
Crystallization temperature (°C)	550	550	520

## Data Availability

Data are contained within the article.

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
