# Peer review of "Enhanced Magnetoimpedance Effect in Co-Based Micron Composite CoFeNiSiB Ribbon Strips Coated by Carbon and FeCoGa Nanofilms for Sensing Applications"

_sensors, 2024, doi:10.3390/s24102961_

Round 1

Reviewer 1 Report

Comments and Suggestions for Authors

The paper “Enhanced magnetoimpedance effect in Co-based micron composite ribbon strips coated by carbon and FeCoGa nanofilms for sensing applications” by Zhen Yang, Jingyuan Chen, Mengyu Liu, Xuecheng Sun, Chong Lei, Yuanwei Shen, Zhenbao Wang, Mengjiao Zhu, and Ziqin Meng describes interesting phenomena in material science. This challenge deals with magnetic properties as remanence and coercivity which shows appreciable applications in our life.
The structures were fabricated according to MEMS technology, where  photolithography was used followed by wet etching and magnetron sputtering. The materials were characterized by XRD, SEM images and AFM measurements. The electrical properties of these devices were evaluated by hysteresis loops and frequency dependencies of MI.
All experimental procedure are thoroughly described and discussed. This paper might be of general interest.

Author Response

Thank you very much for this important comment.

Reviewer 2 Report

Comments and Suggestions for Authors

The authors report the enhanced magnetoimpedance effect in Co-based micron composite ribbon strips. The micron composite ribbon strips were obtained by coating carbon and FeCoGa nanofilms on the surface of quenched ribbon strip via MEMS technology. The effect of carbon and FeCoGa nanofilm coatings on the crystal structure, surface morphology, magnetic properties and magnetoimpedance (MI) effect of composite ribbon strips were systematically investigated. The influence of the carbon layer thickness on coercivity, magnetic anisotropy field, frequency, and maximum MI effect of the composite ribbons were deeply investigated.

In addition, the mechanism of the effect of FeCoGa and carbon nanofilms on the magnetic properties of the composite ribbon was analyzed. Overall, it is an informative and interesting paper with a theoretical basis for improving the sensitivity of MI sensors. The manuscript is well organized, and it can be fluently read by general audience. Some minor questions and suggestions should be properly addressed before the manuscript can be considered for its publication in journal:

(1) Please make clear the materials used to form composite ribbon, both in the title and abstract.

(2) Concerning to MI results and their explanation, the values of MI are not quite large as comparing with those reported in other materials of some hundreds. Maybe to select other amorphous ribbon with very low value of magnetostriction can be more interesting?

(3) The paper contains some grammatical problems and formatting errors, so please proofread it carefully before publication.

(4) In the experiment section, please check carefully the detection range of the applied magnetic field and the AC frequency.

(5)Related with the FeCoGa coated layer, how is the improvement of MI? This layer I imagine is of crystalline nature and lower resistivity and it is reasonable the AC flows easier than for the amorphous ribbon. In addition, this layer exert an additional pressure that can affect the scenarious for MI.

Author Response

(1) Please make clear the materials used to form composite ribbon, both in the title and abstract.

Reply: Thank you very much for this important comment.

The “ribbon” in the title and abstract were revised for “CoFeNiSiB ribbon”, and the nominal composition of commercial Co-based ribbons was Co82.6Fe4.2Ni1.2Si8.8B3.2 and were illustrates in the section of “Materials and Methods”.

(2) Concerning to MI results and their explanation, the values of MI are not quite large as comparing with those reported in other materials of some hundreds. Maybe to select other amorphous ribbon with very low value of magnetostriction can be more interesting?

Reply: Thank you very much for this important comment.

Among various magnetoimpedance (MI) materials, Co-based amorphous ribbons has attracted considerable attention due to their higher permeability, lower saturation field, lower power dissipation, lower hysteresis, and tunable magnetic anisotropy. In addition, they are not sensitive to mechanical stress and can work in harsh environments for a long time, so the stability and reliability of product devices are greatly improved, especially for military products sensing applications. In addition, a higher temperature annealing treatment on the Co-based ribbons will not lead to a certain degree of brittleness compared with Fe-based ribbon.

Compared the adopted ribbon in previous reports, the amorphous ribbon utilized in this design is not only superior to Metglas and Hebei King Do in some key parameters (such as Curie Temperature, Coercive Force and Crystallization Temperature) but also cheaper and easy to purchase. The physical and magnetic properties of the ribbons are presented in Table 1. In subsequent work, the MI performance can be substantially improved after a certain degree of annealing. Although the MI in our report is not high, the MI can be subsequently increased significantly by treatment due to higher Curie Temperature, and applied to more scenarios.

Table 1. Magnetic and physical properties of three commercial ribbons.

Magnetic & Physical Properties

Brand name

Hebei King Do

Metglas

Jiangxi Dayou

Saturation Induction (T)

>0.55

0.5

0.58

Curie Temperature (â—¦C)

205

200

250

Maximum Permeability (µ)

>1,200,000

1,000,000

>1,200,000

Coercive Force (A/m)

<2.0

2

0.3

Density (g/cm3)

8.5

7.59

7.25

Crystallization Temperature (â—¦C)

550

550

520

(3) The paper contains some grammatical problems and formatting errors, so please proofread it carefully before publication.

Reply: Thank you very much for this important comment.

We've carefully proofread our paper sentence by sentence.

(4) In the experiment section, please check carefully the detection range of the applied magnetic field and the AC frequency.

Reply: Thank you very much for this important comment.

The AC frequency was from 100 Hz to 30 MHz. A Helmholtz coil system is used to create a uniform magnetic field from -140 to 140 Oe, which is along the ribbons’ longitudinal axis.

  • Related with the FeCoGa coated layer, how is the improvement of MI? This layer I imagine is of crystalline nature and lower resistivity and it is reasonable the AC flows easier than for the amorphous ribbon. In addition, this layer exert an additional pressure that can affect the scenarious for MI.

Reply: Thank you very much for this important comment.

Firstly, the FeCoGa nanolayer can further reduce the surface roughness of the composite ribbon and effectively enhance the MI effect. Secondly, the FeCoGa coating with 220 nm thickness has a crystalline structure. As a result, the FeCoGa nanolayer has a lower resistivity compared to the amorphous ribbon. This lower resistivity facilitates the flow of alternating current (AC) through the material, leading to improved MI performance. Finally, the presence of the FeCoGa coated layer can exert additional stress on the composite ribbon. Appropriate stress can improve the magnetic domain structure of the composite ribbon, modulate the magnetic properties and improve the MI effect. Furthermore, the deposition of thicker FeCoGa coatings can result in the generation of significant internal stresses, which may lead to the formation of lattice distortions or cracks in FeCoGa films. This, in turn, can influence the magnetic properties and the magnetic impedance effect. Therefore, based on the experimental analysis, the optimal thickness of FeCoGa nanolayer is 220 nm.
